# Exploring the Role of Symptom Diversity in Facial Basal Cell Carcinoma: Key Insights into Preoperative Quality of Life and Disease Progression

**DOI:** 10.3390/cancers17010138

**Published:** 2025-01-04

**Authors:** Domantas Stundys, Alvija Kučinskaitė, Simona Gervickaitė, Jūratė Grigaitienė, Janina Tutkuvienė, Ligita Jančorienė

**Affiliations:** 1Institute of Clinical Medicine, Faculty of Medicine, Vilnius University, 03101 Vilnius, Lithuania; jurate.grigaitiene@gmail.com (J.G.); ligita.jancoriene@santa.lt (L.J.); 2Faculty of Medicine, Vilnius University, 03101 Vilnius, Lithuania; 3Department of Anatomy, Histology and Anthropology, Faculty of Medicine, Vilnius University, 03101 Vilnius, Lithuaniajanina.tutkuviene@mf.vu.lt (J.T.)

**Keywords:** skin cancer, basal cell carcinoma, facial surgery, quality of life, skin cancer index, clinical symptoms, histopathology

## Abstract

Basal cell carcinoma is the most common skin cancer, especially on the face, yet many patients delay seeking medical care despite experiencing symptoms. This study investigates how different symptoms, such as discomfort, pain, or visible tumor presence, impact patients’ quality of life and their decision to consult a doctor. The findings reveal that symptoms significantly affect emotional and social well-being but often do not prompt timely medical attention. On average, patients delayed seeking care for almost two years, which increases disease complexity and the need for more extensive treatment. This research underscores the importance of raising awareness about the symptoms and earlier diagnosis of basal cell carcinoma to improve patient outcomes and reduce the burden on healthcare systems.

## 1. Introduction

Facial basal cell carcinoma (BCC) is the most common form of skin cancer, characterized by slow growth and potential for significant local tissue destruction [1,2]. Despite the serious nature of BCC, much of the existing research on treatment delays has focused on melanoma, leaving a critical gap in understanding how BCC patients behave in similar circumstances [3,4,5].

Although the face is an aesthetically sensitive area, delays in seeking care for BCC remain, reflecting a multifaceted interaction of clinical, psychological, and social factors that require a deeper investigation. A crucial factor in postponing treatment is the patient’s initial decision to seek medical attention [6]. Although the connection between BCC symptoms and their negative impact on quality of life (QoL) is well established, research shows that symptoms alone often fail to prompt timely medical consultations, with many patients delaying care due to denial, neglect, or fear of diagnosis and treatment [7,8,9,10,11].

Furthermore, symptoms such as itch and pain in BCC may be linked to tumor depth and the body’s inflammatory response, potentially signaling disease progression [12]. Despite these indicators, many patients do not recognize the seriousness of the disease, further delaying necessary treatment. As the disease advances, the resulting tissue damage necessitates more complex reconstruction methods, impacting disease burden, patient QoL, and increasing healthcare costs [13,14,15,16,17,18,19]. This highlights the need for greater public awareness about BCC’s signs and symptoms to promote earlier intervention.

This study seeks to explore the relationship between symptom diversity and patient behavior with facial BCC, focusing on how different symptom profiles influence disease-specific preoperative QoL, disease progression, and the timing of the first medical consultation. It also offers a novel viewpoint into how the variety of symptoms influences both clinical outcomes and the overall patient experience.

## 2. Materials and Methods

This study was conducted under the approval of the Vilnius Regional Biomedical Research Ethics Committee (No. 2022/11-1476-943, issued 18 November 2022). In compliance with the Declaration of Helsinki, all study participants provided written informed consent before inclusion. Data collection took place from 23 November 2022 to 19 April 2024 at the Vilnius University Hospital Santaros Klinikos (VUH) Centre of Dermatovenereology in Lithuania.

This cross-sectional analysis was a part of a larger study on nonmelanoma skin cancer that involved 300 cutaneous neoplasms of the face. For the present study, only the data from patients with postoperatively confirmed BCC were included, resulting in a total of 278 consecutive patients. The inclusion criteria focused on adult patients with either a clinically suspected or histopathologically confirmed diagnosis of facial BCC who were presenting for surgery, such as excision, skin-plasty, or skin transplantation. Patients were excluded if they had undergone any facial surgical treatment within the year prior to enrollment, or if they had significant cognitive impairments that could affect data accuracy.

On the day of surgery, demographic data such as age and gender were collected alongside tumor characteristics, including tumor size and histopathological classification into subtypes (superficial, nodular/micronodular, infiltrative, or other). Patients were also asked to report the presence of specific symptoms. These symptoms included pain, described as a sharp or throbbing sensation; bleeding, which could be spontaneous or provoked; itching, defined as persistent or episodic pruritus; tumor presence, referring to a visible or palpable mass; discomfort, characterized as tightness, pressure, or mild aching distinct from pain; and erosion, defined as the breakdown or exposure of underlying skin layers, possibly accompanied by oozing. QoL was assessed using the Skin Cancer Index (SCI), which evaluates emotional, social, and appearance domains. Patients were also asked whether the tumor caused anxiety, which was analyzed separately from the physical symptoms and SCI scores.

### Statistical Analysis

All statistical analyses were conducted using Python 3.12.5, and statistical significance was set at *p* < 0.05.

Differences in categorized tumor size, symptom count, and time until the first doctor appointment across age and gender groups were analyzed using chi-square tests. Additionally, chi-square tests were used to examine whether a particular gender, in relation to tumor size and age group, sought medical care sooner or later.

To determine whether QoL scores varied based on the presence of different symptoms, one-way ANCOVA calculations for each SCI subscale were conducted. Additionally, multiple linear regression analyses were performed to further investigate the individual effects of symptoms on QoL. The dependent variables in these analyses were the SCI subscales: emotional, social, appearance, and total SCI scores. The independent variables included categorical indicators for the presence of specific symptoms, such as Discomfort, Tumor, Pain, Itching, Erosion, and Bleeding. Each model estimated the unique contribution of each symptom to the QoL outcomes while controlling other symptoms. Interaction terms, chosen based on theoretical justification and clinical relevance, were included to explore potential combined effects of symptoms on QoL. The overall significance of each regression model was evaluated using ANOVA with the F-test, assessing whether the predictors collectively explained a significant portion of the variance in QoL outcomes.

To investigate the association between multiple symptoms and histological types, we performed separate multivariate logistic regression analyses for each histological type. The symptoms included as predictors were Discomfort, Tumor, Pain, Itching, Erosion, and Bleeding.

The influence of each symptom and their interactions on the likelihood of seeking medical care was analyzed using multiple logistic regression models, focusing on the time frames of ≤6 and >6 months, as well as ≤12 and >12 months. Additionally, a Cox Proportional Hazards model was employed to assess the association between symptoms and the time until patients sought medical care. The proportional hazards assumption was tested to validate the model. Hazard ratios, along with their confidence intervals, were interpreted to determine whether the presence of a symptom was associated with a shorter or longer time to seek medical care.

## 3. Results

### 3.1. Descriptive Statistics

The study analyzed data from 278 participants, of whom 65% were women and 35% were men. Most BCC cases were observed in patients aged 65–74 (27%) and 75–84 (25%), while younger patients (<44 years) accounted for only 5% of the cohort. Notably, 207 patients (74%) presented with a single histological type of BCC, and 71 patients (26%) had collision tumors. Both sexes had equal representation in superficial BCC (50%), but women demonstrated a higher prevalence in nodular (65%) and infiltrative BCC (66%) compared to males (35% and 34%, respectively). The average tumor size was 10.07 mm, with no significant differences in tumor size observed across gender or age groups (*p* > 0.05) (Table 1).

Patients reported a diverse range of symptoms, with tumor presence (27%) being the most frequently observed, followed by erosion (18%) and discomfort (17%). Bleeding, itching, and pain were less commonly reported, accounting for 13%, 10%, and 2% of cases, respectively. Most participants (46%) reported more than three symptoms at presentation, while 44% reported two to three symptoms, and only 9% reported a single symptom. Reported symptom characteristics by histologic tumor type, time from the symptom onset, and symptom specific QoL are presented in Table 2 and Table 3.

The mean time from the symptom onset to the first doctor appointment was 21 months. The results highlighted higher BCC rates in older age groups and females, with nodular BCC being the most common subtype. Neither tumor size, nor the number of symptom presence, or the time until first doctor appointment differed by sex or age (*p* > 0.05). Additionally, there was no clear tendency for either men or women, based on their age or tumor size, to seek medical care earlier or later (categorized as <6 months vs. ≥6 months or <12 months vs. ≥12 months) (*p* > 0.05).

### 3.2. Impact of Symptoms on Quality of Life

The ANCOVA results indicated that patients with tumor-associated local discomfort had statistically significantly lower QoL scores in all three SCI subscales: emotional (F = 6.55, *p* = 0.011), social (F = 5.35, *p* = 0.022), appearance (F = 4.06, *p* = 0.045), and total (F = 7.69, *p* = 0.006) (Appendix A). The multiple regression analyses revealed that discomfort was also a statistically significant factor negatively impacting QoL in all domains: (β = −1.96, *p* = 0.011), social (β = −1.00, *p* = 0.022), appearance (β = −0.76, *p* = 0.045), and total (β = −3.71, *p* = 0.006) (Appendix A). The relatively low R-squared values for the models suggested that, while those symptoms did affect QoL, there were likely additional factors influencing QoL that were not captured in this analysis. In contrast, symptoms such as pain, bleeding, itching, tumor presence, and erosion did not significantly influence QoL scores. Interaction effects between symptoms were also non-significant, suggesting that their combined presence does not amplify their individual impact on QoL.

### 3.3. Symptom Associations with Histological Subtypes

Three patients presented with BCC types other than infiltrative, superficial, or nodular, and were thus excluded from the analysis. Multivariate logistic regression models were applied to explore the association between various symptoms and different histopathological types. The analysis revealed that bleeding was statistically less common in superficial BCC (β = −0.93, *p* = 0.033), while infiltrative BCC was associated with a lower likelihood of palpable tumor presence (β = −1.21, *p* = 0.005). Erosion showed a marginal association with infiltrative BCC (β = 0.59, *p* = 0.067). No significant symptom associations were observed for nodular BCC, indicating that this subtype may present with fewer distinctive symptoms. The coefficients for nodular BCC indicated weak relationships between symptoms and the histological type, such as discomfort (β = 0.21, *p* = 0.498), tumor (β = 0.23, *p* = 0.640), pain (β = −0.21, *p* = 0.696), itching (β = 0.11, *p* = 0.753), erosion (β = 0.38, *p* = 0.300), and bleeding (β = −0.57, *p* = 0.122), none of which reached statistical significance (Appendix A).

### 3.4. Care-Seeking Behaviors

#### 3.4.1. Likelihood to Seek Medical Care Within 6 Months

None of the individual symptoms significantly influenced the likelihood of seeking medical care within 6 months. The model’s pseudo R-squared value was relatively low, indicating that while the model explained some variance in the outcome, other factors not included in this analysis might also play an important role in determining when patients seek care. The analysis suggested that neither the individual symptoms alone, nor the combination of them significantly influenced early care-seeking behavior (Appendix A).

#### 3.4.2. Likelihood to Seek Medical Care Within 12 Months

The analysis showed that the presence of a tumor (coefficient: −5.3836, *p* = 0.040) and pain (coefficient: −6.3793, *p* = 0.031) significantly increased the likelihood of seeking medical care within 12 months. In contrast, itching, discomfort, anxiety, erosion, and bleeding all showed statistically insignificant effects on care-seeking behavior. The interaction between tumor and itching was significant, indicating that patients with both symptoms were more likely to seek care over a longer time frame (*p* = 0.022) (Appendix A).

#### 3.4.3. Association Between Symptoms and Time to Seeking Medical Care

The results of Cox regression analysis overall suggest limited evidence for strong predictive value among the symptoms, with no covariate reaching conventional significance levels (*p* < 0.05). However, anxiety showed a borderline association with a shorter time to seek care (HR = 1.24, *p* = 0.08) (Table 4).

## 4. Discussion

Consistent with the global BCC trends, our study cohort reflects significant demographic and clinical patterns [20]. The results reveal a significant prevalence of BCC in individuals aged 65 and older, highlighting the correlation between age and increased skin cancer risk due to cumulative sun exposure and other environmental factors [21,22,23].

It might seem reasonable to assume that more severe symptoms would negatively impact QoL, motivating patients to seek help sooner. However, our findings challenge this notion. Our analysis showed that only tumor-associated local discomfort was significantly linked to a decline in QoL, affecting emotional, social, and appearance aspects of patients’ lives. Interestingly, symptoms like bleeding, erosion, and the presence of a tumor itself did not significantly impact QoL. In contrast, Gaulin et al. found that symptoms such as pain and discomfort notably detracted from patients’ daily lives [24]. This aligns with findings from Gordon et al., who noted pain and discomfort to be the most frequently reported issues, followed by anxiety and depression [25].

In this study, we also investigated the association between symptoms and different histopathological types of basal cell carcinoma (BCC), focusing on the infiltrative, superficial, and nodular subtypes. Our findings indicated that certain symptoms might be linked to the specific BCC types. Notably, we observed that bleeding was statistically less common in patients with superficial BCC (β = −0.93, *p* = 0.033). This aligns with existing literature suggesting that superficial BCCs, characterized by their less aggressive nature, often present as scaly patches and may less likely disrupt local blood vessels compared to other subtypes [26]. Additionally, our analysis revealed that patients with infiltrative BCC were less likely to report a palpable tumor (β = −1.21, *p* = 0.005), which may reflect the diffuse growth pattern of this subtype that often leads to less distinct tumor formation [26]. In contrast, our study found no statistically significant associations between symptoms and nodular BCC, indicating that this subtype may present with a different symptom profile. The weak relationships observed for symptoms such as discomfort, pain, and bleeding in nodular BCC (e.g., discomfort β = 0.21, *p* = 0.498; pain β = −0.21, *p* = 0.696) suggest that this subtype may often be asymptomatic until it reaches a significant size. These findings highlight the value of understanding the unique clinical characteristics of different histopathological types of BCC. Although they do not influence the timely care, they aid in guiding the final diagnosis.

A study by Yosipovitch et al. highlights that while pain and itch are not frequent symptoms in melanoma, they are more prevalent in nonmelanoma skin cancers, suggesting a potential correlation between these symptoms and the histological characteristics of the tumors [12]. This association may be attributed to the inflammatory response elicited by the tumor and its interaction with surrounding tissues, which can lead to pain and discomfort.

In examining care-seeking behavior for BCC, individual symptoms alone did not significantly influence early medical attention within 6 months, highlighting unmeasured factors in patient decision-making. Over a 12-month period, pronounced symptoms like tumor presence and pain significantly increased care-seeking likelihood, while symptoms such as itching, discomfort, and bleeding lacked urgency. Notably, the interaction between tumor presence and itching amplified care-seeking behavior, suggesting the importance of combined symptom effects.

Cox Regression analysis further revealed no significant associations between most symptoms and time to care-seeking, with hazard ratios close to one. However, a trend towards earlier care was observed with anxiety, hinting at a potential role of psychological factors. These findings emphasize the complexity of symptom perception in driving timely intervention for skin cancer.

The mean time of 21 months from symptom onset to the first medical appointment which we have observed in our study is alarming. Moreover, the lack of significant differences in tumor size, symptom presence, or time to first appointment based on sex or age raises important questions about the factors influencing patient behavior in seeking medical care. This delay is also echoed in the literature, which often cites denial of illness, older age, and difficulty scheduling doctor‘s appointment as significant factors contributing to late diagnosis of skin cancers, including BCC [6,11].

Despite the significant impact that discomfort and related symptoms have on their daily lives, many patients delay seeking medical advice, which can lead to the progression of the disease and more complex treatment requirements. This delay in seeking care is particularly concerning given that no individual symptom, including bleeding, pain, or itching, consistently drives patients to seek earlier medical attention. This underscores the need for a more proactive approach in the management of facial BCC. Healthcare providers should not only focus on treating the visible signs of the condition but also on understanding and addressing the underlying discomfort that patients experience. This could involve more comprehensive assessments during routine check-ups, where patients are encouraged to discuss any discomfort or emotional distress they may be feeling, even if they do not perceive these symptoms as severe. Moreover, this situation highlights a broader issue in patient behavior: the tendency to underreport or ignore symptoms that are not perceived as immediately threatening. This can be due to a variety of factors, including fear, denial, or a lack of awareness about the seriousness of the condition. To counteract this, it is essential to improve patient counseling and education. Healthcare providers should emphasize the importance of early medical evaluation, regardless of the presence or severity of specific symptoms, to prevent the potential worsening of the condition. In practice, this could mean implementing more structured follow-up schedules, providing patients with detailed information about what to watch for, and creating an open line of communication where patients feel comfortable reporting changes in their condition, no matter how minor they may seem. Additionally, involving patients in their care through shared decision-making could increase their engagement and likelihood of seeking timely medical advice. Ultimately, addressing these issues requires a multi-faceted approach that includes not only medical treatment but also patient education, psychological support, and a healthcare system that prioritizes early intervention. By doing so, we can improve the overall quality of life for patients with facial BCC and potentially reduce the burden of this condition on both individuals and the healthcare system.

### 4.1. Strengths

This study stands out as one of the few to link symptoms with histological tumor types in nonmelanoma skin cancer, providing a unique perspective on the interplay between clinical presentation and tumor characteristics. With a large sample size of 278 patients, it captures a broad and representative spectrum of BCC demographics, aligning with global trends and enhancing the generalizability of the findings. The detailed data collection on symptomatic, demographic, and behavioral factors, combined with rigorous statistical methods, offers valuable insights into the relationship between symptom diversity, quality of life, and care-seeking behavior.

### 4.2. Limitations

The cross-sectional design restricts the ability to observe changes in symptoms, QoL, and patient behavior over time. Potential recall bias in self-reported symptoms could slightly affect data accuracy. Additionally, as the study was conducted within a single institution, the findings may not fully capture variations across different populations or healthcare systems.

## 5. Conclusions

Discomfort is the primary factor leading to a decline in QoL in patients with facial BCC. They postpone seeking medical advice, leaving lesions averaging 1 cm in size untreated for nearly two years. Given that no single symptom significantly influenced patients to seek earlier medical attention, it is essential to improve patient counseling, stressing the importance of early medical evaluation, irrespective of specific symptoms. This highlights the pressing need for greater awareness and more effective skin cancer prevention strategies.

## Figures and Tables

**Table 1 cancers-17-00138-t001:** Baseline demographic and clinical characteristics of the enrolled patients.

Demographic Characteristics		Verified BCC Histologic Types				
	Total Patients, *n* = 278	Total Verified BCC Types, *n* = 353	Superficial BCC, *n* = 40	Nodular BCC, *n* = 223	Infiltrative BCC, *n* = 87	Other Types of BCC, *n* = 3
**Sex**						
Male	97 (35%)	129 (37%)	20 (50%)	78 (35%)	30 (34%)	1 (33%)
Female	181 (65%)	224 (63%)	20 (50%)	145 (65%)	57 (66%)	2 (67%)
**Age, years (%)**						
<44	15 (5%)	19 (5%)	3 (8%)	13 (6%)	3 (3%)	0
45–54	36 (13%)	41 (12%)	6 (15%)	28 (13%)	7 (8%)	0
55–64	52 (19%)	64 (18%)	3 (8%)	44 (20%)	17 (20%)	0
65–74	76 (27%)	94 (27%)	11 (28%)	61 (27%)	22 (25%)	0
75–84	70 (25%)	93 (26%)	11 (28%)	57 (26%)	23 (26%)	2 (67%)
>85	29 (10%)	42 (12%)	6 (15%)	20 (9%)	15 (17%)	1 (33%)
**Largest tumor diameter (mean), mm**		10.07	12.13	9.98	10.89	9.33
**Number of symptoms**	**Total patients, *n* = 278**		**Superficial BCC, *n* = 40**	**Nodular BCC, *n* = 223**	**Infiltrative BCC, *n* = 87**	**Other types of BCC, *n* = 3**
None	1 (0%)		0	0	1 (1%)	0
1 symptom	26 (9%)		4 (10%)	20 (9%)	6 (7%)	0
2–3 symptoms	122 (44%)		19 (48%)	101 (45%)	33 (38%)	2 (67%)
>3 symptoms	129 (46%)		17 (43%)	102 (46%)	47 (54%)	1 (33%)

**Table 2 cancers-17-00138-t002:** Patient symptom characteristics by neoplasm type and time from symptom onset.

Symptoms		Reported Number of Symptoms by Histologic Tumor Types, *n* = 923				Time from the Symptom Onset to 1st Visit
		Superficial BCC, *n* = 121	Nodular BCC, *n* = 741	Infiltrative BCC, *n* = 308	Other Types of BCC, *n* = 9	
Discomfort	160 (17%)	22 (18%)	130 (18%)	56 (18%)	3 (33%)	22.96
Anxiety	120 (13%)	20 (17%)	97 (13%)	41 (13%)	1 (11%)	18.7
Tumor presence	252 (27%)	35 (29%)	203 (27%)	72 (23%)	3 (33%)	20.76
Pain	22 (2%)	1 (1%)	17 (2%)	8 (3%)	0	24.95
Itching	89 (10%)	13 (11%)	72 (10%)	27 (9%)	1 (11%)	22.86
Erosion	162 (18%)	20 (17%)	131 (18%)	60 (19%)	0	23.68
Bleeding	118 (13%)	10 (8%)	91 (12%)	44 (14%)	1 (11%)	25.08

**Table 3 cancers-17-00138-t003:** Patient symptom-specific quality of life per symptom.

Symptom	Total Symptoms, *n* = 923	SCI Score			
		SCI Emotional	SCI Social	SCI Appearance	SCI Total
Discomfort	160 (17%)	25.92	21.54	11.98	59.44
Anxiety	120 (13%)	25.5	21.4	11.79	58.69
Tumor presence	252 (27%)	26.90	22.09	12.42	61.41
Pain	22 (2%)	25.27	21.41	11.54	58.23
Itching	89 (10%)	26.11	21.65	12.38	60.15
Erosion	162 (18%)	27.05	22.18	12.38	61.55
Bleeding	118 (13%)	26.94	22.03	12.30	61.28

**Table 4 cancers-17-00138-t004:** Cox proportional hazards model results: association between symptoms and time to seeking medical care.

Covariate	Coefficient (coef)	Exp (Coefficient)	SE (coef)	Coef Lower 95%	Coef Upper 95%	Exp (Coefficient) Lower 95%	Exp (Coefficient) Upper 95%	Z-Value	*p*-Value (*p*)	-Log2(P)
Discomfort	−0.128	0.88	0.127	−0.377	0.121	0.686	1.129	−1.006	0.314	1.669
Tumor	0.169	1.184	0.221	−0.265	0.603	0.767	1.828	0.764	0.445	1.169
Pain	−0.1	0.905	0.229	−0.548	0.348	0.578	1.416	−0.438	0.661	0.597
Itching	−0.06	0.942	0.135	−0.325	0.206	0.722	1.228	−0.441	0.659	0.6
Erosion	−0.185	0.831	0.143	−0.466	0.095	0.628	1.099	−1.296	0.195	2.358
Bleeding	−0.181	0.834	0.144	−0.462	0.1	0.63	1.106	−1.261	0.207	2.269
Anxiety	0.219	1.244	0.126	−0.029	0.466	0.971	1.594	1.73	0.084	3.581

## Data Availability

Data analyzed during this study are not publicly available but are available from the corresponding author upon reasonable request.

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
