# Peer review of "Exploring the Role of Symptom Diversity in Facial Basal Cell Carcinoma: Key Insights into Preoperative Quality of Life and Disease Progression"

_cancers, 2025, doi:10.3390/cancers17010138_

Round 1

Reviewer 1 Report

Comments and Suggestions for Authors

The role of symptom diversity in facial basal cell carcinoma was investigated by the authors.

278 patients were included with confirmed BCC biological types, n= 353 nodular BCC and infiltrative BCC being the most common type with a mean tumour size of 1 cm.

This is quite large for a facial site and suggests a significant delay in primary diagnosis.

The authors conducted a thorough statistical analysis of quality of life based on the presence of different symptoms combined with multiple regression analysis. However, the cross-sectional design did not allow changes over time to be observed.

The authors convincingly demonstrate the delay in diagnosis in an elderly population with subjective and obvious clinical symptoms. Only logistic regression showed that tumour presence and pain were associated with earlier care-seeking within 12 months (p < 0.05).

This paper makes a strong case for skin cancer prevention. I recommend that this statement be strengthened as a final statement of this paper.

Author Response

We thank the Reviewer for their thoughtful and constructive comments on our manuscript. Below, we address each of the points raised.

Comment 1: "This is quite large for a facial site and suggests a significant delay in primary diagnosis."

Response 1: We appreciate your observation. We have clarified in the discussion section that a notable delay in diagnosis exists as elderly patients do not consider other symptoms significant while the tumor reaches substantial size of around 1 cm, which aligns with the patterns observed in our elderly population.

Comment 2: The authors conducted a thorough statistical analysis of quality of life based on the presence of different symptoms combined with multiple regression analysis. However, the cross-sectional design did not allow changes over time to be observed.

Response 2: We appreciate your recognition of our statistical analysis and the use of multiple regression to explore the impact of symptom diversity on quality of life. We acknowledge that the cross-sectional design limits the ability to observe changes over time in limitations section. Future studies with a longitudinal design are planned to address this limitation and provide deeper insights into how symptom progression and quality of life evolve over time.

Comment 3: "This paper makes a strong case for skin cancer prevention. I recommend that this statement be strengthened as a final statement of this paper."

Response 3: We agree with your suggestion. To address this, we have revised the conclusion (lines 305-306, 308-310) to emphasize the critical need for enhanced skin cancer prevention strategies, including public awareness and education campaigns.

We believe these changes have strengthened the manuscript and better reflect the valuable feedback provided. Thank you again for your insightful comments.

Reviewer 2 Report

Comments and Suggestions for Authors

This paper describes a study about the role of symptom diversity in facial basal cell carcinoma. The paper makes a statistical analysis of data from the primary factors leading to a decline in QoL in patients with facial BCC. The work is based on a significant sample. More factors could be analyzed but that is out of the scope of the preset paper. Probably the interactions of the factors under analysis could be research to check what are the main factors associated with "discomfort". 

Author Response

Thank you for your thoughtful review and valuable feedback. I appreciate your recognition of the study's focus on symptom diversity in facial BCC and its impact on quality of life (QoL).

This study is still ongoing, which is why we haven’t considered additional variables yet. However, I agree that analyzing more factors and their interactions could provide valuable insights into patient discomfort, and these concerns will be addressed in future stages. Thank you again for your thoughtful input!

Reviewer 3 Report

Comments and Suggestions for Authors

The work by Stundys D. et al represents a detailed mathematical analysis of  the relationship between the symptoms of  facial basal cell carcinoma, quality of life (QoL) of the patients and care-seeking behaviors. A total of 278 patients of Vilnius University Hospital from November 2022 to April 2024 with histologically confirmed BCC were analyzed focusing on the impact of symptoms on clinical outcomes and consultation timing. Multiple linear regression analyses with dependent variables corresponding to SCI (Skin Cancer Index) scores like emotional, social, appearance investigated the individual effects of symptoms on QoL.while multivariable logistic regression analysis for each histological type assessed the impact of symptoms on care-seeking behavior.

Although the work itself cannot be called outstanding but it points attention to the fact that almost 2 years (21 month) is a mean time from the symptoms onset to the fist seeking for the medical help and thus stimulate a healthcare system to pay attention not only on treatment of patients but also to educate them explaining the advantage of early cancer diagnostics. The article is useful and should be published after small technical corrections of the Tables which make the article more comfortable for the readers. It is desirable that the title of the columns and their contents in tables 1 and 2 be on the same page and the word “discomfort” in Table 3 be on one line instead of “dicomfor  t”.

Author Response

Thank you for your thoughtful feedback and constructive suggestions. We appreciate your acknowledgment of the study’s importance in highlighting the significant delay in seeking medical help for facial BCC and the need for increased education on the benefits of early cancer diagnosis. Regarding the technical corrections, we will ensure that the tables are adjusted for clarity and readability, including aligning the column titles and contents on the same page and correcting the formatting of the word “discomfort” in Table 3.